# Distinct molecular pathways mediate Mycn and Myc-regulated miR-17-92 microRNA action in Feingold syndrome mouse models

Fatemeh Mirzamohammadi[1], Anastasia Kozlova[1], Garyfallia Papaioannou[1], Elena Paltrinieri[1], Ugur M. Ayturk[2] & Tatsuya Kobayashi[1]

Feingold syndrome is a skeletal dysplasia caused by loss-of-function mutations of either *MYCN* (type 1) or *MIR17HG* that encodes miR-17-92 microRNAs (type 2). Since miR-17-92 expression is transcriptionally regulated by MYC transcription factors, it has been postulated that Feingold syndrome type 1 and 2 may be caused by a common molecular mechanism. Here we show that *Mir17-92* deficiency upregulates TGF-β signaling, whereas *Mycn*-deficiency downregulates PI3K signaling in limb mesenchymal cells. Genetic or pharmacological inhibition of TGF-β signaling efficiently rescues the skeletal defects caused by *Mir17-92* deficiency, suggesting that upregulation of TGF-β signaling is responsible for the skeletal defect of Feingold syndrome type 2. By contrast, the skeletal phenotype of *Mycn*-deficiency is partially rescued by *Pten* heterozygosity, but not by TGF-β inhibition. These results strongly suggest that despite the phenotypical similarity, distinct molecular mechanisms underlie the pathoetiology for Feingold syndrome type 1 and 2.

[1] Endocrine Unit, Massachusetts General Hospital, and Harvard Medical School, Boston 02114 MA, USA. [2] Musculoskeletal Integrity Program, Hospital for Special Surgery, New York 10021 NY, USA. Correspondence and requests for materials should be addressed to T.K. (email: tkobayashi1@mgh.harvard.edu)

Heterozygous mutations in *MYCN* or *MIR17HG* in humans cause Feingold syndrome that is characterized by skeletal developmental defects including microcephaly, short stature, and brachysyndactyly with diminished middle phalanxes[1–3]. *MYCN*, a member of the *MYC* family proto-oncogenes, encodes a transcription factor that regulates genes that promote cell growth and proliferation[4]. Previous studies have shown that MYC (Myc in mice) transcription factors bind directly to the *MIR17HG* gene (*Mirc1*, also known as *Mir17-92* in mice) to stimulate the expression of miR-17-92 microRNAs (miRNAs) encoded by this gene[5, 6].

miR-17-92 cluster miRNAs are oncogenic miRNAs that have been studied mainly in the context of cancers[7, 8]. miR-17-92 miRNAs regulate genes involved in cell cycle and apoptosis, such as *Bim* (*Bcl11*)[9, 10] and *E2f1*[5], as well as target genes encoding signaling molecules[9, 11–13] in a context-dependent manner to promote tumorigenesis.

miR-17-92 miRNA family miRNAs, comprised of four functionally overlapping groups of miRNAs, are also encoded by two additional paralogous genes, *MIR106B* (*Mirc3*, also known as *Mir106b-25* in mice) and *MIR106A* (*Mirc2*, also known as *Mir106a-363* in mice). The physiologic role of miR-17-92 family miRNAs during animal development was demonstrated by genetic mutant mice missing these genes[10]. Homozygous deletion of *Mir17-92* in mice impairs animal growth and causes prenatal lethality. Additional deletion of *Mir106b-25* results in significantly more severe developmental defects, providing genetic evidence for the overlapping function of these genes.

Based on the fact that deletion of either *MYCN* or *Mir17HG* results in similar skeletal defects in humans and that miR-17-92 expression is regulated by Myc transcription factors, it has been hypothesized that Mycn and miR-17-92 miRNAs function in the same pathway to regulate skeletal development[2]. To test this hypothesis, we conditionally deleted *Mir17-92* with or without deletion of *Mir106b-25* in the developing limb bud and skull mesenchyme.

In this study, we present evidence that Mycn and miR-17-92 miRNAs regulate skeletal progenitor cell proliferation through distinct signaling pathways. We show that overactivation of TGF-β signaling causes the skeletal defects of miR-17-92 miRNA-deficient limbs and the skull, whereas downregulation of PI3K/Akt signaling is a major contributor to the skeletal phenotype caused by *Mycn*-deficiency. This study using in vivo models also suggests that physiological effects of miRNAs and transcription factors, which regulate multiple genes, converge onto relatively limited signaling pathways.

## Results

**Generation of mouse models of Feingold syndrome patients**. In order to determine physiologic roles of miR-17-92 miRNAs in skeletal development, we deleted *Mir17-92* with or without its paralogous gene, *Mir106b-25*, in skeletal mesenchymal cells using *Prx1-Cre* transgenic mice and floxed *Mir17-92* and *Mir106b-25*-null mice. Mice with heterozygous *Mir17-92* deletion (*Prx1-Cre: Mir17-92*[fl/+]) exhibited modest skeletal defects similar to those in germline heterozygous null mice[2] (Supplementary Fig. 1a-d). The skeletal phenotype was more pronounced after deletion of both alleles of *Mir17-92* (Fig. 1; Supplementary Fig. 1a-d). Simultaneous deletion of *Mir17-92* and *Mir106b-25* (doubly conditional knockout, *Prx1-Cre:Mir17-92*[fl/fl]:*Mir106b-25*[−/−], *17* dKO) caused microcephaly, cutaneous syndactyly, and brachydactyly, phenotypes similar to those observed in patients with Feingold syndrome type 2 (Fig. 1a–d). While heterozygous *Mycn* germline deletion in mice does not cause detectable skeletal abnormalities[14], conditional deletion of *Mycn* in developing limbs (*Prx1-*

*Cre:Mycn*[fl/fl], *Mycn* cKO) resulted in severe brachysyndactyly similar to that observed in patients with Feingold syndrome type 1 and also in mice previously reported[15] (Fig. 1a–d).

*Mycn*-deficient limb mesenchymal cells isolated from embryos at indicated embryonic days (E) showed significant reductions in miR-17-92 miRNA levels over time, as assessed by quantitative RT-PCR (qRT-PCR) and in situ hybridization (Fig. 1e; Supplementary Fig. 2). Thus, we attempted to rescue the skeletal phenotype of *Mycn*-deficient limbs by overexpressing miR-17-92 miRNAs using Cre-inducible *Mir17-92* transgenic mice[9]. *Mir17-92* overexpression in transgenic mice (*Prx1-Cre:Mir17-92*[Tg], *17* Tg) caused limb overgrowth (Supplementary Fig. 3), and it partially rescued the skeletal abnormalities of *Mycn* cKO mice (Fig. 1f). These results are consistent with the notion that the action of Mycn is mediated, at least in part, by miR-17-92 miRNAs in skeletal development.

**Proliferation defects in miR-17-92 miRNA-deficient cells**. To assess the biological consequence of loss of *Mir17-92* and *Mir106b-25* in the skeletal mesenchyme, we measured cell proliferation and apoptosis in the skull and limbs of mutant embryos. Simultaneous loss of *Mir17-92* and *Mir106b-25* in the early limb bud and skull mesenchyme caused a significant decrease in cell proliferation (Fig. 2a–c), which was also confirmed in vitro (Fig. 2d). We did not find significant changes in cell death assessed by the TUNEL assay (Supplementary Fig. 4a). The domain expressing *Sox9*, a marker of precartilaginous condensations, was reduced in size in *17* dKO mice at E9.5 and E11.5, presumably due to the proliferation defect of *17* dKO limb mesenchymal cells (Fig. 2e). Conditional deletion of *Mir17-92* and *Mir106b-25* exclusively in the cells of the precartilaginous condensation and their descendants using *Col2a1-Cre* transgenic mice caused no obvious skeletal abnormalities (Supplementary Fig. 4b), suggesting that the proliferation defect in cells of the limb bud mesenchyme, but not cells in the precartilaginous condensation, is responsible for the skeletal abnormalities of *17* dKO mice. Since *Mycn* deficiency also reduces limb bud cell proliferation[15], these findings suggest that similar cellular mechanisms underlie the pathogenesis of Feingold syndrome type 1 and type 2.

**TGF-β and STAT3 signaling upregulation in *Mir17-92*-deficiency**. Although a miRNA potentially regulates hundreds of target genes, it is proposed that miRNA–RNA interactions are evolutionarily selected to cooperatively regulate certain biological pathways[16–18]. In addition to directly regulating genes involved in cell cycle and apoptosis[5, 9], miR-17-92 miRNAs are reported to regulate genes encoding signaling molecules in a context-dependent manner to promote tumorigenesis[11, 13, 19]. Therefore, we first determined the status of major signaling pathways in primary limb or skull mesenchymal cells missing *Mir17-92* and *Mir106b-25*. At the basal culture condition, we found upregulation in TGF-β and JAK/STAT3 signaling (Fig. 3a, b). In addition, *Mir17-92:Mir106b-25*-deficient cells showed a greater response in Smad2 phosphorylation than control after acute stimulation with TGF-β1 (Fig. 3c). The type II TGF-β receptor, Tgfbr2, a predicted target of miR-17 and miR-19 group miRNAs[19], was upregulated at both the protein and mRNA levels (Fig. 3a, d). Since loss of miR-17 group miRNAs (miR-17 and miR-20a) causes skeletal defects in mice[20], we tested whether miR-17 can suppress gene expression via the predicted target sequence of *Tgfbr2* by luciferase reporter assay. miR-17 co-transfection significantly reduced the expression of a luciferase reporter construct with the wildtype, but not with a mutated, binding sequence (Fig. 3e). Although *Bim* and *Pten* were previously identified as critical miR-

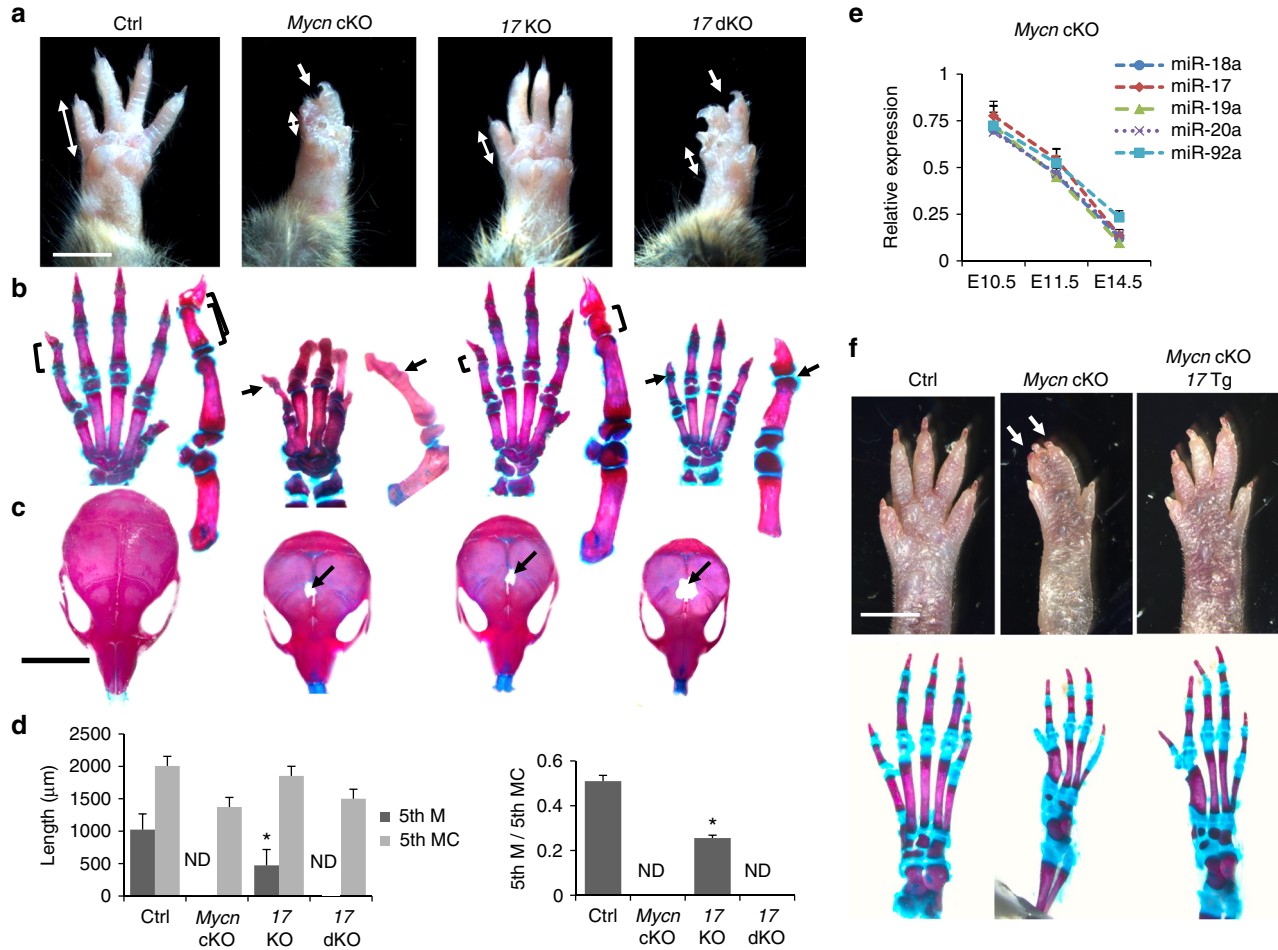

**Fig. 1** Generation of mouse models of Feingold syndrome. **a** Forelimbs of wildtype control (Ctrl), *Mycn* Knockout (*Prx1-Cre:Mycn^fl/fl*, *Mycn* cKO), *Mir17-92* knockout (*Prx1-Cre:Mir17-92^fl/fl*, *17* KO), and *Mir17-92* and *Mir106b-25* doubly conditional knockout (*Prx1-Cre:Mir17-92^fl/fl:Mir106b-25^−/−*, *17* dKO) mice at postnatal day (P) 16. *17* KO mutants exhibit mild brachydactyly in the fifth digit (double-headed arrows). Cutaneous syndactyly (arrows) and severe brachydactyly are presented in *17* dKO and *Mycn* cKO mice. Scale bar, 0.5 cm. **b** Alizarin red and alcian blue staining of the forelimb and magnified views of the fifth digit of corresponding mice in **a**. *17* KO mutants show shortening of the middle phalanx (M) (brackets) of the fifth digit. *17* dKO and *Mycn* cKO mutants show severe brachymesophalangy and the absence of the middle phalanx of the fifth digit (arrows). **c** Alizarin red and alcian blue staining of the skull of corresponding mice in **a**. *Mycn* cKO, *17* KO, and *17* dKO mice show microcephaly and a frontal bone ossification defect (arrows). Scale bar, 1 cm. **d** Quantification of the length of the fifth M and fifth metacarpal bone (5th MC) and the ratio of 5th M to 5th MC. Values are expressed as mean ± SE (*n* = 5 in each group, *\*p* < 0.001 vs. Ctrl). ND, not detected. **e** Relative expression of five miRNAs encoded in the *Mir17-92* gene in limb bud mesenchymal cells isolated from *Mycn* cKO embryos at embryonic day 14.5. (relative to control littermate samples, *n* = 5 each group, *<0.001 vs. Ctrl at every time point). **f** Hindlimbs of wildtype Ctrl, *Mycn* cKO, and *Mycn* Knockout overexpressing *Mir17-92* (*Prx1-Cre:Mycn^fl/fl:Mir17-92^Tg*, *Mycn* cKO:*17* Tg) (upper panels) and alizarin red and alcian blue staining of the hindlimbs of corresponding genotypes (bottom panels). *Mir17-92* overexpression partially rescues the skeletal defects of *Mycn* cKO mutants. Cutaneous syndactyly (white arrows) and shortening of the fifth digit in *Mycn* cKO mutants are rescued by overexpression of *Mir17-92* whereas fused fourth and fifth metatarsal bones still remain (bottom panels). Scale bar, 0.5 cm

17-92 targets in lymphogenesis and tumorigenesis[9, 10], we did not find upregulation in *Bim* or *Pten* in *17* dKO mesenchymal cells (Fig. 3d), supporting the notion that the regulatory role of miR-17-92 miRNAs is dependent on the cellular context.

**Causal role of TGF-β signaling in cell proliferation defect**. To test whether the upregulation of the TGF-β or JAK/STAT3 signaling was responsible for the proliferation defect observed in *17* dKO cells, we inhibited these signaling pathways using pathway-specific inhibitors in vitro. Whereas treatment with the STAT3 inhibitor, S31-201, failed to promote proliferation of *17* dKO cells in vitro (Fig. 4a, b), treatment with the TGF-β receptor inhibitor, LY364947, significantly ameliorated the proliferation defect of *17* dKO cells (Fig. 4c, d). This finding was

confirmed by experiments using GW788388, another TGF-β receptor inhibitor, and also using the neutralizing antibody against TGF-β ligands, 1D11 (Supplementary Fig. 5a, b). These results suggest that the TGF-β upregulation is responsible for the proliferation defect, a finding consistent with the previous reports that overactivation of TGF-β signaling reduces limb mesenchymal cell proliferation and inhibits limb growth[21, 22].

**TGF-β inhibition rescues skeletal defects of *17* dKO mice**. We treated *17* dKO mice with TGF-β receptor inhibitors from E9.5 through postnatal day (P) 16.5. *17* dKO mice treated with LY364947 showed significant normalization of limb and skull development (Fig. 4e–h). Moreover, genetic suppression of TGF-β signaling via conditionally deleting one allele of *Tgfbr2* in *17*

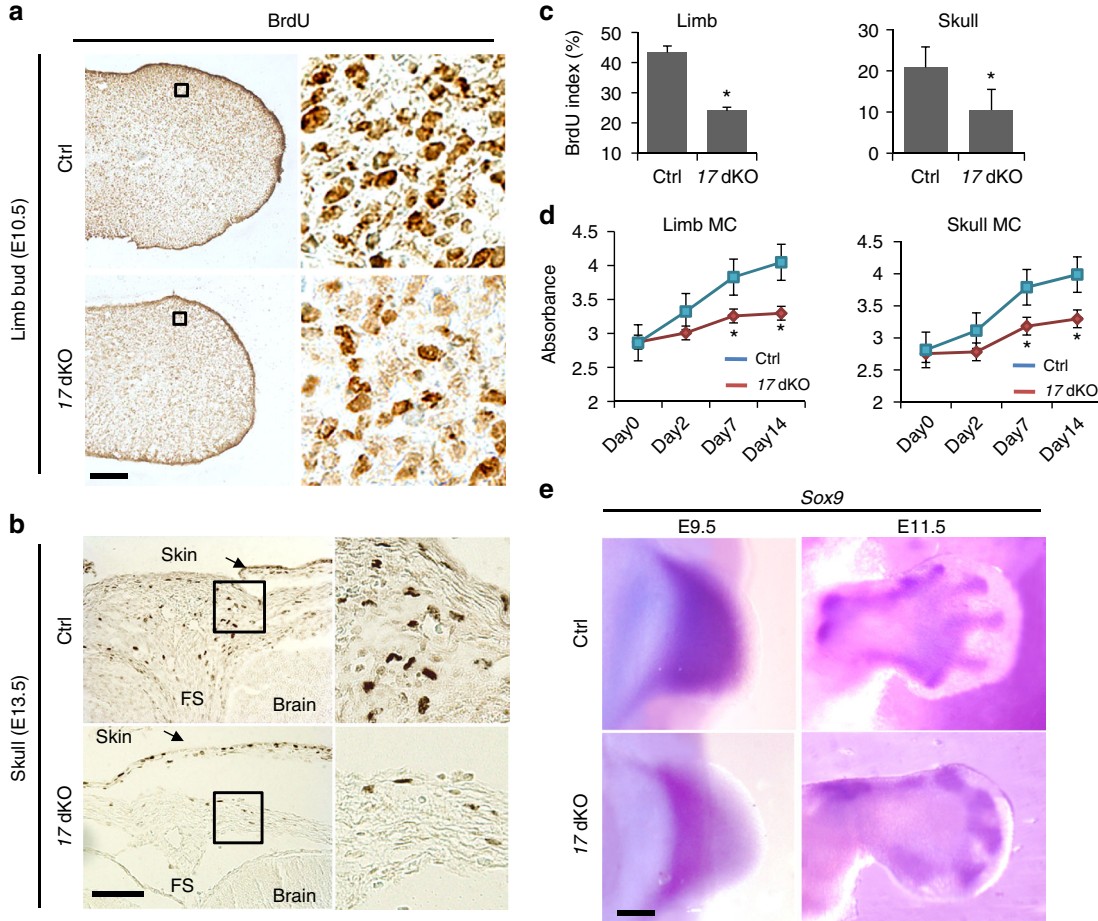

**Fig. 2** Proliferation defect in cells missing *Mir17-92* and *Mir106b-25*. **a–d** Loss of *Mir17-92* and *Mir106b-25* (*17* dKO) impairs skeletal mesenchymal cell proliferation. Since *Mir106b-25*-null mice show no skeletal abnormalities, *Mir106b*-null mice were used as the control. **a** BrdU labeling on limb bud sections of *Mir17-92^{fl/fl}:Mir106b-25^{−/−}* (Ctrl) and *Prx1-Cre:Mir17-92^{fl/fl}:Mir106b-25^{−/−}* (*17* dKO) embryos at embryonic day (E) 10.5. **b** BrdU labeling of coronal skull sections of Ctrl and *17* dKO embryos at E13.5. **c** The BrdU-positive cells were significantly reduced in the *17* dKO limb bud and skull mesenchyme (*n* = 6, *p* < 0.05). **d** Cell proliferation assay on limb bud and skull mesenchymal cells (MC) in vitro. Primary mesenchymal fibroblasts isolated from *Mir17-92^{fl/fl}: Mir106b-25^{−/−}* embryos were transduced with adenoviruses expressing a Cre recombinase (*17* dKO) or a yellow fluorescent protein (Ctrl) to delete *Mir17-92* in vitro. The efficient reduction (85 ± 5%) in miR-17-92 miRNAs was confirmed by qRT-PCR. *Mir17-92*-deficient limb bud and calvarial mesenchymal cells showed significantly impaired proliferation over time compared with control, as assessed by an index of the increase in cell mass (*n* = 5, *p* < 0.05). **e** Whole-mount in situ hybridization for *Sox9* shows a reduction in size of the *Sox9*-positive domain at E9.5 and E11.5. FS frontal suture. Scale bar, 100 μm

dKO mice (*17* dKO:*Tgfbr2* Het) also partially rescued the skeletal abnormalities of *17* dKO mice (Supplementary Fig. 5c). A short-term treatment regimen during early stages of limb development (E9.5–E15.5) with TGF-β receptor inhibitor mostly rescued the limb and skull abnormalities whereas treatment during late embryonic stages (E15.5–E18.5) showed no effect (Supplementary Fig. 5d). These data suggest that miR-17-92 miRNA-mediated suppression of TGF-β signaling through *Tgfbr2* downregulation is particularly important during early stages of skeletal development and that the TGF-β signaling upregulation is responsible for skeletal defects in Feingold syndrome type 2. TGF-β inhibitor treatment also rescued the digit phenotype of mice with heterozygous *Mir17-92* deletion, a genetic alteration close to that of patients with Feingold syndrome type 2, further supporting this notion (Supplementary Fig. 6a).

**Distinct molecular mechanisms in *17* dKO and *Mycn* cKO mice**. In contrast to *17* dKO mice, suppression of TGF-β signaling via *Tgfbr2* heterozygous deletion or TGF-β receptor inhibitor treatment failed to rescue the skeletal defects of *Mycn* cKO mice (Fig. 5a, b; Supplementary Fig. 6b). Consistent with this result, we

did not find upregulation in TGF-β signaling in *Mycn*-deficient cells, as indicated by the unaltered level of p-Smad2 (Fig. 5c, d). These results conflict with the previously proposed model in which miR-17-92 miRNAs mediate Mycn's action[2], and are also inconsistent with our earlier observations that miR-17-92 expression is reduced in E14.5 *Mycn*-deficient limbs and that miR-17-92 overexpression rescues the *Mycn* cKO digit phenotype (Fig. 1e, f). However, we noted that the reduction of miR-17-92 miRNA expression levels in *Mycn* cKO cells was relatively modest at E10.5 when TGF-β suppression by miR-17-92 miRNAs is critical to maintain limb mesenchymal cell proliferation (Fig. 1e). Consistent with this finding, known miR-17-92 miRNAs target genes, including *Tgfbr2*, were not deregulated in *Mycn*-deficient cells at E10.5 (Fig. 5d). We compared the gene expression profiles of *Mycn*-deficient and *17* dKO limb mesenchymal cells by RNA-Seq, but found an only limited overlap in genes whose expression was altered in these two groups (Supplementary Fig. 7a; Supplementary Data 1). Together, these data demonstrate that, unlike in the model of Feingold syndrome type 2, TGF-β deregulation does not play a causal role in the pathogenesis of the type 1 Feingold syndrome model (*Mycn* deletion).

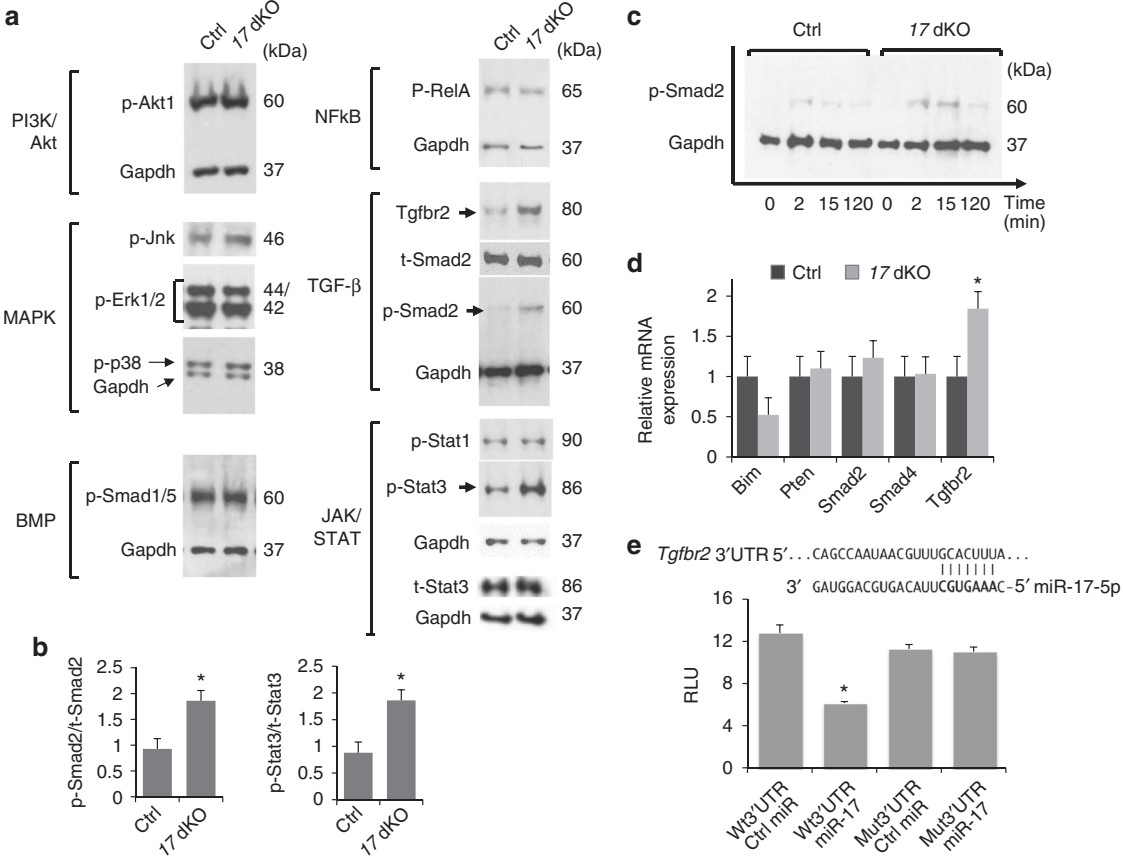

**Fig. 3** TGF-β and STAT3 upregulation in miR-17-92 miRNA-deficient cells. **a** Immunoblot analysis on indicated proteins using protein lysates from cells cultured in DMEM containing 10% FBS. Levels of p-Smad2, Tgfbr2, and p-Stat3 (arrows) are increased in *Mir17-92:Mir106b-25*-deficient limb bud cells (*17* dKO). **b** Quantification of the p-Smad2 to total Smad2 (t-Smad2) and p-Stat3 to total Stat3 (t-Stat3) protein ratios based on western blot data (*n* = 3, \**p* < 0.05). **c** Increased TGF-β signaling in *17 dKO* cells upon stimulation. Serum-starved cells were treated with 50 ng/ml TGF-β1. Smad2 phosphorylation was analyzed at the indicated time points. **d** Relative mRNA expression of known targets of miR-17-92 miRNAs in *17 dKO* limb bud cells (*n* = 3, \**p* < 0.05). **e** Luciferase reporter assay for miR-17 regulation on *Tgfbr2* binding site. Primary limb bud cells were co-transfected with control miRNA mimic (Ctrl miR) or mmu-miR-17-5p (miR-17) and a luciferase reporter construct carrying a wildtype (Wt 3′UTR) or mutated 3′UTR (Mut 3′UTR) sequence of mouse *Tgfbr2*. The predicted binding sequence of *Tgfbr2* 3′UTR and relative luciferase units (RLU) are shown (*n* = 6, \**p* < 0.001)

**Role of PI3K downregulation in *Mycn* cKO mice.** To further investigate the molecular mechanism for the type 1 model, we revisited the fact that miR-17-92 overexpression rescued the *Mycn* cKO digit phenotype. The miR-17-92 miRNA levels in the limb mesenchymal cells of *17* Tg mice were four times more abundant than those in control (Supplementary Fig. 8a). We also found that these supraphysiological levels of miR-17-92 miRNAs suppressed the expression of *Pten*, a negative regulator of phosphoinositide 3-kinase (PI3K)/Akt signaling, and upregulated the phospho-Akt (p-Akt) level (Supplementary Fig. 8b, c). Thus, we hypothesized that deregulation of the PI3K/Akt signaling pathway was involved in the pathogenesis of the *Mycn* cKO phenotype. Indeed, we found that *Mycn*-deficient mesenchymal cells showed a decreased level of p-Akt (Thr308), suggesting downregulation of the PI3K signaling pathway (Fig. 5c).

To test whether the suppression of the PI3K signaling pathway is responsible for the skeletal abnormalities in *Mycn* cKO, we deleted one allele of *Pten* in *Mycn* cKO mice (*Mycn* cKO:*Pten* Het). Although *Pten* haploinsufficiency itself did not cause any skeletal phenotype, we found that *Pten* heterozygous deletion in *Mycn*-deficient skeletal progenitor cells partially rescued the skeletal abnormality of *Mycn* cKO mice to a similar extent to *Mir17-92* overexpression (*Mycn* cKO: *17* Tg) (Fig. 6a, b). We confirmed that the p-Akt level increased in *Mycn* cKO limb bud cells after deletion of one allele of *Pten* (Fig. 6c). In contrast, *Pten*

heterozygous deletion showed no effect on the *17* dKO skeletal phenotype (Supplementary Fig. 8d). These data suggest that downregulation of the PI3K signaling pathway plays a major pathogenic role in the Feingold syndrome type 1, but not type 2.

Regarding the mechanism by which *Mycn* deficiency reduces PI3K signaling, there were no significant changes in the expression level of genes related to the PI3K complex, *Pten*, or their downstream effectors in *Mycn* cKO limb bud cells (Supplementary Fig. 7b). However, we found decreased phosphorylation of the tail region of Pten (p-Pten) that negatively regulates Pten activity[23] (Fig. 6d). Casein kinase 2 (CK2) is a major regulator of Pten phosphorylation and activity[24–26], and CK2 and Myc have been shown to synergistically promote oncogenesis via the CK2's promoting action on Myc stabilization[27]. Therefore, we examined the expression and activation of CK2. While the total expression level of CK2 genes was unaltered (Supplementary Fig. 7c, d), CK2 phosphorylation, which reflects CK2 activity[28], was reduced in *Mycn*-deficient cells (Fig. 6e). These results suggest that the downregulation of the CK2/Pten cascade is responsible, at least in part, for the reduced PI3K signaling caused by *Mycn* deficiency.

## Discussion

In this study, we generated mouse models that exhibited skeletal phenotypes similar to those in patients with Feingold syndrome

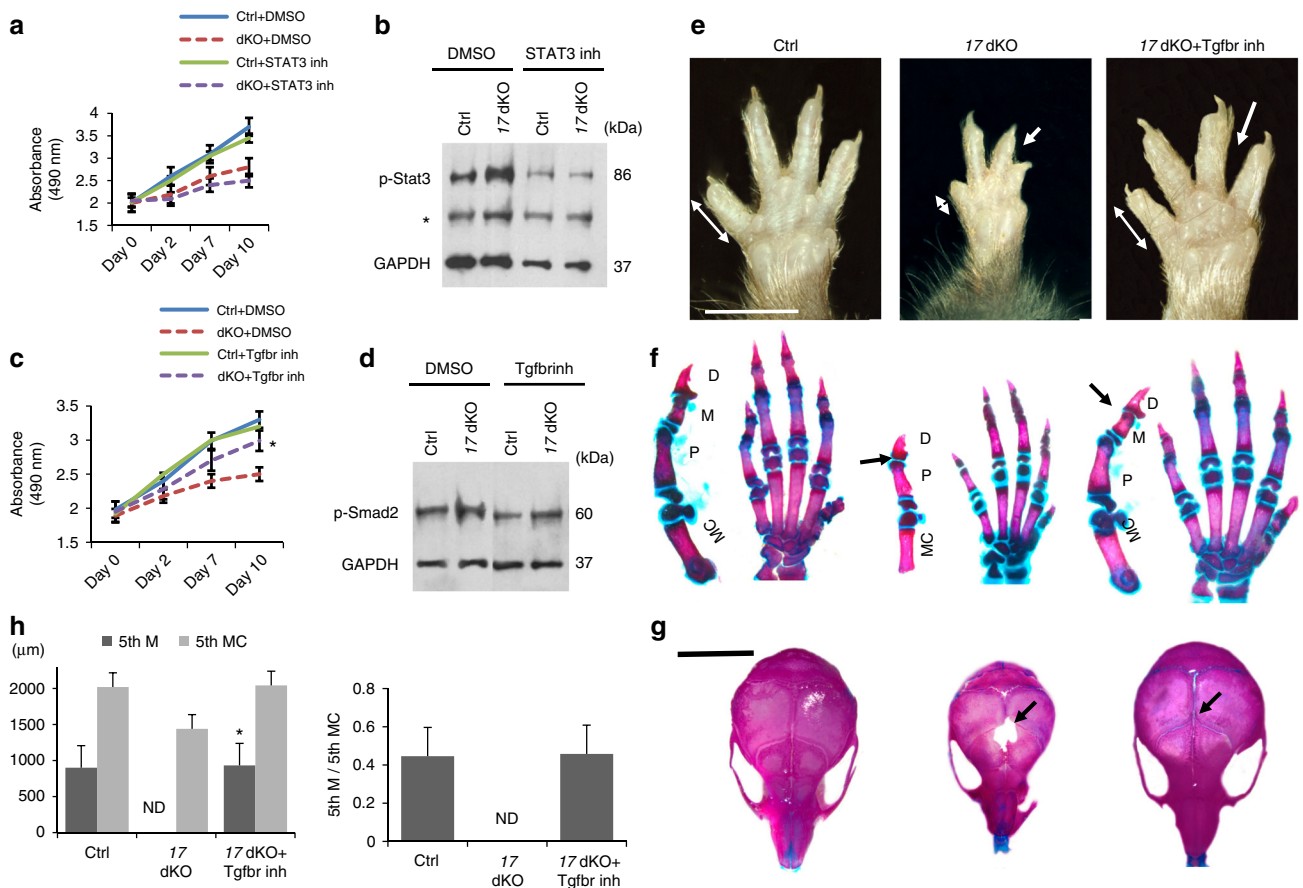

**Fig. 4** Rescue of cell proliferation defects with TGF-β receptor inhibitors. **a** Cell proliferation assay on control (Ctrl) and *Mir17-92:Mir106b-25*-deficient (*17* dKO) limb bud mesenchymal cells treated with vehicle (DMSO) or a STAT3 inhibitor (STAT3 inh, S31-201, 100 μM). Ctrl and *17* dKO limb bud cells were prepared by transducing YPF (Ctrl) or Cre (*17* dKO) in vitro. STAT3 inhibitor treatment shows no significant improvement in *17* dKO cell proliferation. **b** Treatment of limb bud cells with S31-201 downregulates the p-Stat3 level; *non-specific band. **c** Cell proliferation assay on Ctrl and *17* dKO (dKO) cells treated with DMSO or a TGF-β receptor inhibitor (Tgfbr inh, Ly364947, 0.2 μM). Treatment with Ly364947 has no significant effect on control cells whereas it significantly ameliorates the proliferation defect of *17* dKO cells ($n = 6$, *$p < 0.05$ vs. *17* dKO + DMSO). **d** Treatment of limb bud cells with Ly364947 downregulates the p-Smad2 level. **e**–**g** Ly364947 treatment efficiently rescues the skeletal defects of *17* dKO mutants. Ly364947 (1 mg/kg/day, i.p.) was injected into pregnant and nursing mothers from E9.5 through P7.5, and then injected directly into individual mice. The shortening of the fifth digit (double arrows) syndactyly (white arrows) (**e**), missing middle phalanx (M) (**f**), and microcephaly and frontal bone ossification defect (black arrows) (**g**) in *17* dKO mutants were ameliorated by Ly364947 treatment. **h** Quantification of the length of the fifth mesophalanx (5th M), fifth metacarpal bone (5th MC), and the ratio of 5th M to 5th MC. Values are expressed as mean ± SE ($n = 6$ each group, $p < 0.001$ vs. dKO). Scale bars: 0.5 cm in **e**, 1.0 cm in **g**. D distal phalanx; P proximal phalanx

type 1 and type 2 by conditionally deleting *Mycn* and *Mir17-92*, respectively, in skeletal progenitors. Our findings strongly suggest that TGF-β signaling upregulation in mesenchymal progenitor cells plays a major causal role in the skeletal abnormalities of Feingold syndrome type 2, whereas downregulation of the PI3K signaling pathway significantly contributes to the pathogenesis of Feingold syndrome type 1 (Fig. 6f).

Although these mouse models are not exact genetic equivalents of human patients with heterozygous deletions of these genes, considering that heterozygous germline-null mice do not fully recapitulate the phenotypes of patients with Feingold syndrome type 1 and type 2 and that our mouse models recapitulate well the skeletal abnormalities of patients, our mouse models likely share the same pathophysiological mechanisms of these diseases in humans. In addition, the finding that TGF-β inhibition was able to rescue the digit defect (shortening of the middle phalanx) of heterozygous *Mir17-92* conditional deletion also supports that the TGF-β deregulation underlies the pathogenesis of Feingold syndrome type 2.

Considering that the cellular mechanism that accounts for the skeletal defects, i.e., reduced proliferation of mesenchymal progenitors, appears to be common in *Mir17-92:106b-25*-deficient (present study) and *Mycn*-deficient limbs and skulls[15] and that the well-established molecular relationship between Myc transcription factors and miR-17-92 miRNAs, our finding that deficiencies of Mycn and miR-17-92 miRNAs causes aberrations in distinct signaling pathways was unexpected. As previous study showed[2], we also found partial reductions in miR-17-92 miRNAs in *Mycn*-deficient limbs. Considering that *Mycn*-deficient limbs show a skeletal phenotype more severe than that of *Mir17-92:106b-25* doubly conditional nullizygotes, the partial reduction of miR17-92 miRNAs in *Mycn*-deficient limbs may have only limited biological effects.

In the mouse model for Feingold syndrome type 2, using genetic and pharmacological interventions, we have demonstrated that the upregulation in TGF-β signaling due to derepression of *Tgfbr2*, a direct target of multiple miR-17-92 miRNAs, plays the causal role. The fact that suppression of

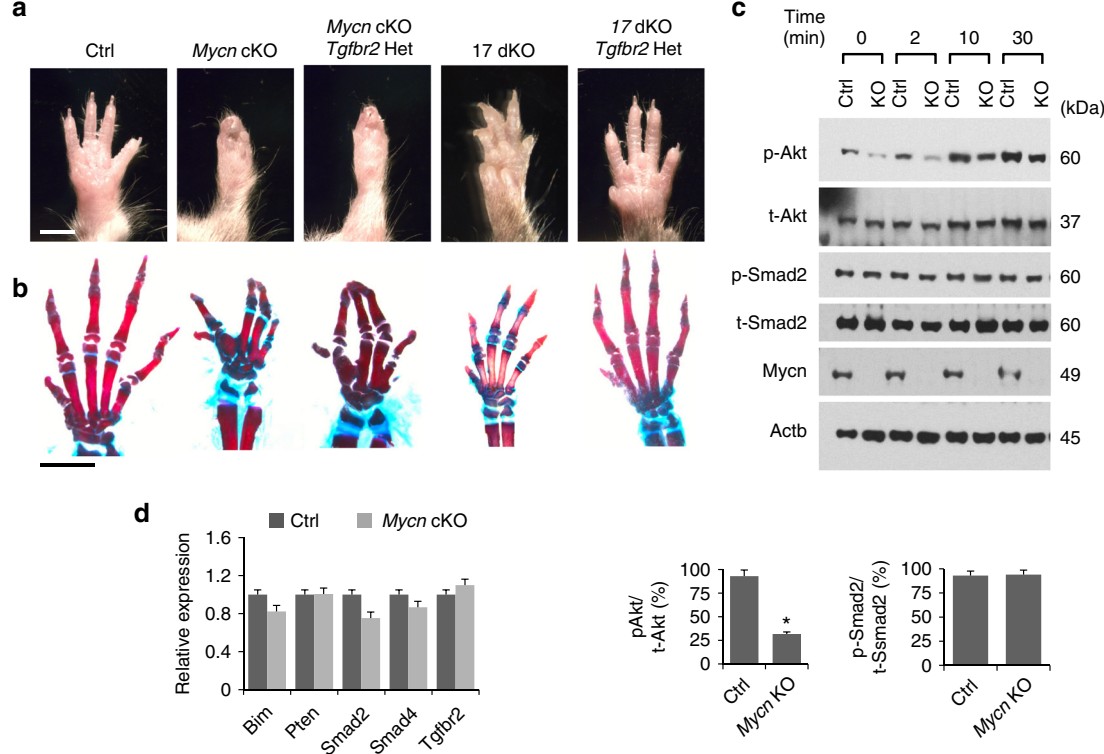

**Fig. 5** Distinct mechanisms in Feingold syndrome mouse models. **a** Forelimbs of mice with indicated genotypes at postnatal day 16. Wildtype control (Ctrl), *Mycn* conditional Knockout (*Prx1-Cre:Mycn*^fl/fl^, *Mycn* cKO), *Mycn* cKO in which one allele of *Tgfbr2* is deleted (*Prx1-Cre:Mycn*^fl/fl^*:Tgfbr2*^fl/+^, *Mycn* cKO:*Tgfbr2* Het), *Mir17-92* and *Mir106b-25* doubly conditional knockout (*Prx1-Cre:Mir17-92*^fl/fl^*:Mir106b-25*^−/−^, *17* dKO), and *17* dKO in which one allele of *Tgfbr2* is deleted (*Prx1-Cre:Mir17-92*^fl/fl^*:Mir106b-25*^−/−^*:Tgfbr2* ^fl/+^, *17* dKO:*Tgfbr2* Het). Unlike *17* dKO mice, the digit abnormalities of *Mycn* cKO mutants are not rescued by *Tgfbr2* heterozygous deletion. **b** Alizarin red and alcian blue staining of forelimbs of corresponding mice in **a**. Scale bars, 0.5 cm. More than three rescued mice per each model were analyzed to confirm the reproducibility. **c** Immunoblot analysis for indicated proteins upon treatment with 10% fetal bovine serum (FBS) at indicated time points (top panel). Primary mesenchymal fibroblasts from limb buds of E10.5 *Mycn*^fl/fl^ embryos were transduced with adenoviruses expressing a Cre recombinase (KO) or a yellow fluorescent protein (Ctrl) to delete *Mycn* in vitro. Cells were serum starved for 1 h before stimulation. The efficient reduction in the Mycn protein level was confirmed. Levels of p-Akt (Thr308) and p-S6k are decreased in *Mycn*-deficient limb bud cells (KO). Quantification of p-Akt relative to total Akt (t-Akt) and p-Smad2 to total Smad2 (t-Smad2) at time 0 based on the western blot data (bottom panels) (n = 3, *p < 0.05). **d** Relative expression of previously reported targets of miR-17-92 miRNAs in *Mycn* cKO limb bud mesenchymal cells isolated from embryos at age E10.5. Y axis, arbitrary units

TGF-β alone was able to almost completely rescue the skeletal defects of *Mir-17-92:106b-25*-deficient limbs and skulls indicates that *Tgfbr2* is the most physiologically important target of miR-17-92 miRNAs in the context of skeletal development, despite the fact that miR-17-92 miRNAs regulates many other genes in diverse types of cells[29]. Deletion of the endogenous miR-17-92 miRNAs showed no effects on the expression of other known targets, including Pten. However, overexpression of miR-17-92 miRNAs to a supraphysiological level did suppress Pten. This result suggests that suppressive effects of miR-17-92 miRNAs on their targets are dependent on the cellular context, absolute expression level of miRNAs, and perhaps miRNA-target RNA stoichiometry.

We have shown that the downregulation of PI3K/Akt signaling contributes to the skeletal phenotype of *Mycn* cKO mice. Myc transcription factors control cell growth and proliferation downstream of diverse signaling stimuli[30]; however, little is known how Myc transcription factors regulate signaling pathways. We have found that *Mycn*-deficiency decreases Pten phosphorylation and CK2 activation, suggesting the alteration of the CK2/Pten pathway is likely the cause for the reduced PI3K/Akt signaling in *Mycn*-deficient cells. The precise mechanism by which *Mycn* deficiency alters the CK2/Pten pathway is unclear at the moment.

In summary, we have demonstrated that distinct molecular mechanisms are responsible for skeletal phenotypes of mouse models of Feingold syndrome type 1 and type 2. Our results also suggest that targeting specific signaling pathways is a valid therapeutic approach to Feingold syndrome.

## Methods

**Mice**. Floxed *Mir17-92* mice and *Mir106b-25*-null mice[10], Cre-inducible *CAG-Mir17-92* transgenic mice[9], *Prx1Cre* transgenic mice[31], *Col2Cre* transgenic mice[32], and floxed *Mycn* mice[33] were previously described. Mice are in a C57/B6-dominant mixed background. Mutant mice were compared with littermate control mice except mice treated with chemical inhibitors during pregnancy. Since both males and females show the same phenotype with similar degrees, males and females were not discriminated. The mutant mice generally showed qualitative phenotypes; more than three independent experiments were performed to ensure the reproducibility. Due to the nature of experiments, no randomization or blinding was used.

The animal experiments were approved by the Institutional Animal Care and Use Committee (IACUC) of the Massachusetts General Hospital and performed in accordance with the regulations and guidelines.

**Reagents**. Recombinant mouse TGF-β1 (#5231) was purchased from Cell Signaling Technology. Ly364947 (#S2805), GW788388 (#S2750), and S3I-201 (#S1155) were purchased from Selleck Chemicals.

**TGF-β receptor inhibitor injection**. TGF-β receptor inhibitors, Ly364947 and GW788388, were dissolved in DMSO at the concentration of 1 mg/ml and 2 mg/

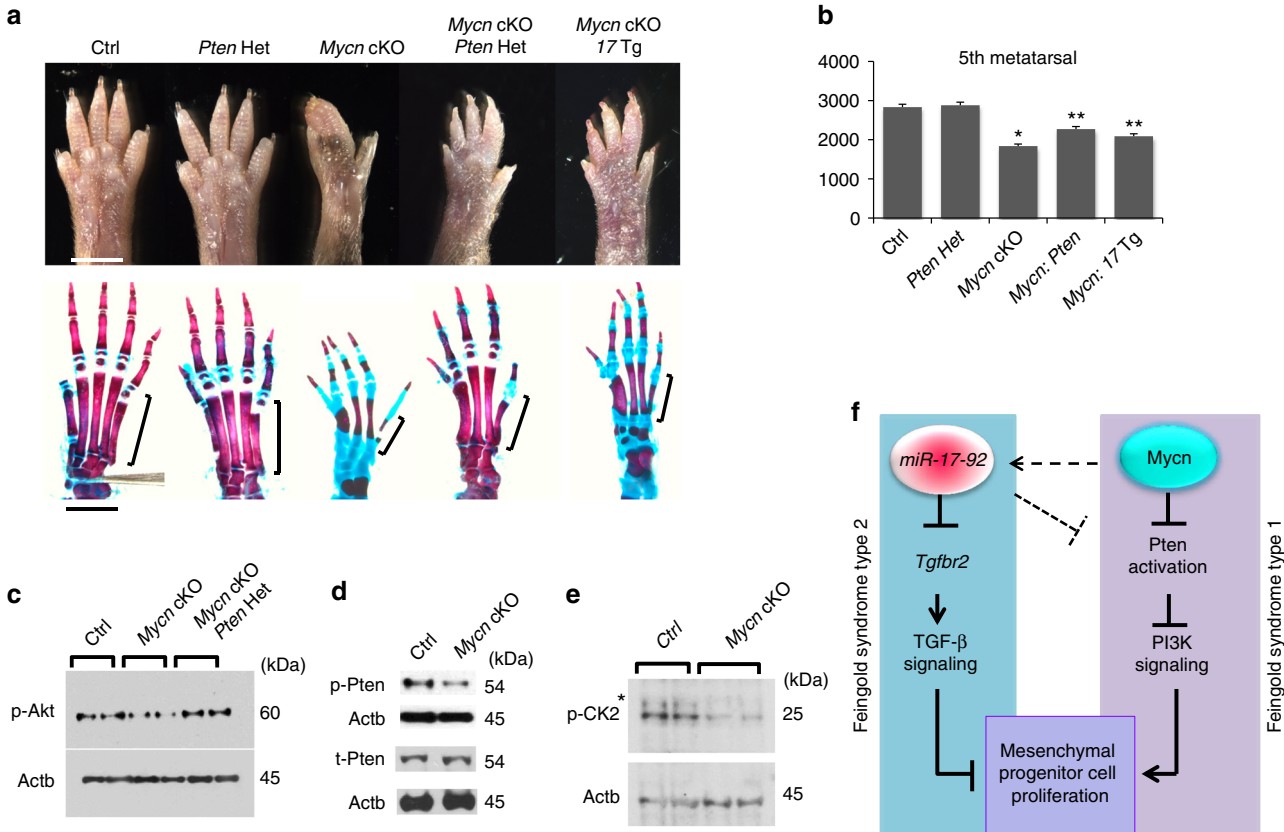

**Fig. 6** PI3K downregulation contributes to the *Mycn* cKO phenotype. **a** Hindlimbs of mice with indicated genotypes at postnatal day 16. Wildtype control (Ctrl), heterozygous *Pten* mutant (*Prx1-Cre:Pten*$^{fl/+}$, *Pten* Het), *Mycn* Knockout (*Prx1-Cre:Mycn*$^{fl/fl}$, *Mycn* cKO), *Mycn* cKO in which one allele of *Pten* is deleted (*Prx1-Cre:Mycn*$^{fl/fl}$:*Pten*$^{fl/+}$, *Mycn* cKO:*Pten* Het), and *Mycn* Knockout overexpressing *Mir17-92* (*Prx1-Cre:Mycn*$^{fl/fl}$:*Mir17-92*$^{Tg}$, *Mycn* cKO:*17* Tg). *Pten* heterozygous deletion partially rescues the skeletal abnormalities of *Mycn* cKO mutants. Cutaneous syndactyly and brachydactyly are partially rescued, while the fused metatarsals and shortening of mid phalanxes remain. The fifth metacarpal bones, subjected to measurement in **b**, are indicated by brackets. Scale bars, 0.5 cm. **b** Quantification of the length (μm) of fifth metacarpal bones of mice with indicated genotypes (*n* = 6 for Ctrl and *Mycn* cKO, *n* = 5 for *Pten* Het and *Mycn* cKO:*Pten* Het, *n* = 4 for *Mycn* cKO:*17* Tg; *$p < 0.01$ vs. Ctrl, **$p < 0.01$ vs. *Mycn* cKO). **c** PI3K signaling was assessed by p-Akt (Thr308) in limb bud cells isolated from E 10.5 mice with indicated genotype. **d** Immunoblot analysis for phospho-Pten (p-Pten) and total Pten (t-Pten) isolated from *Mycn* cKO and wildtype littermate limb bud cells. The p-Pten level is decreased in *Mycn*-deficient skeletal progenitor cells whereas the t-Pten level is unchanged. **e** Immunoblot analysis for casein kinase 2 beta phosphorylation on S209 (p-CK2) in limb bud mesenchymal cells isolated from E10.5 *Mycn* cKO and wildtype littermates. p-CK2 is decreased in *Mycn*-deficient cells; *non-specific band. **f** Proposed model. Upregulation of TGF-β signaling plays a causal role in the skeletal defect of Feingold syndrome type 2 (*Mir17-92* mutation), whereas downregulation of the PI3K signaling plays a major pathophysiologic role in Feingold syndrome type 1 (*Mycn* mutation). Mycn partially regulates miR-17-92 miRNA levels, but the contribution of this regulation in limb development is limited (dotted arrow). Overexpression of miR-17-92 miRNAs can suppress Pten, although miR-17-92 miRNAs have limited regulatory effects on Pten expression at the physiological level (dotted inhibitory line)

ml, respectively. Ly364947 (1 mg/kg/day) and GW788388 (1 mg/kg/day) were injected intraperitoneally (i.p.) into pregnant mothers or nursing mothers until P7.5. Mice older than P7.5 were individually injected.

**Skeletal preparation and histology**. Alizarin red and alcian blue staining was performed using a modified McLeod's method[34].

For histological analysis, the embryos were dissected, fixed in 10% formalin, paraffin-processed, cut, and subjected to BrdU staining and TUNEL staining using the In situ Cell Death Detection kit (Sigma-Aldrich).

**Cell proliferation assays**. For BrdU labeling, BrdU (50 mg/kg) was injected into pregnant mice i.p. 2 h before euthanasia. BrdU was detected using the BrdU in situ staining kit (Invitrogen). The BrdU labeling index was calculated as the ratio of BrdU-positive stained nuclei over total nuclei. In vitro cell proliferation was quantified using the PrestoBlue Cell Viability Reagent (Molecular Probes) according to the manufacturer's instruction.

**Primary limb bud cells isolation and culture**. Isolation and culture of primary limb bud cells of E9.5 through E11.5 mice were performed as previously described[35]. Briefly, embryos were isolated from timed-mated females by cesarean section. Limb buds, dissected using fine tweezers, were treated with 0.25% trypsin containing 2.21 mM EDTA for 5 min to disperse cells before plating. After

overnight culture, cells were trypsinized and replated at the concentration of 5 × $10^5$/ml in a DMEM medium containing 10% fetal bovine serum (FBS). Upon reaching confluence, cells were collected to assess the basal status of indicated signaling pathways. Cells were also serum-starved for 3 h before stimulating with 10% FBS containing 50 ng/ml of TGF-β1. To assess Smad2 phosphorylation, cells were lysed at indicated time points and subjected to Western blot analysis. For RNA isolation, four limb buds of E10.5 and E11.5 embryos were directly lysed in Trizol. For E14.5 limbs, paws of four limbs were dissected to remove the skin and lysed in Trizol.

For in vitro *Mir17-92* deletion, primary limb cells isolated from *Mir17-92* homozygous floxed, *Mir106b* null embryos at E10.5 and were cultured in a DMEM medium containing 10% FBS until reaching 50% confluence. Adenoviruses expressing either Cre or YFP (control) were then added to delete *Mir17-92*. Then cells were split and grown in 96-well plates for indicated days.

**In situ hybridization**. Non-radioactive whole-mount in situ hybridization of mouse embryos was done using a standard protocol[36]. The Sox9 probe was previously described[37]. Section in situ hybridization for miR-17-5p was performed according to the standard protocol with minor modifications[38]. E14.5 forelimbs of *Prx1-Cre:Mycn*$^{fl/fl}$ mice and *Cre*-negative littermate control were formalin-fixed, paraffin-processed, cut at 5 μm thickness. The DIG-labeled miR-17 oligo probe (mmu-miR-17-5p; YD00615470-BCG) was purchased from Qiagen.

**Adenovirus production and infection**. Adenoviruses expressing Cre or YFP were gifts from Dr. Murat Bastepe (Harvard Medical School). Amounts of adenoviruses used for infection were empirically determined to achieve >75% infection efficiency. The infection efficiency was determined 2 days after infection by visualizing YFP for YFP-expressing adenoviruses or by quantifying floxed gene products for Cre-expressing adenoviruses.

**Primary embryonic limb bud cell isolation**. $Prx1Cre: Mycn^{fl/+}$ male and $Mycn^{fl/fl}$ female mice were mated and mating was confirmed by the presence of a vaginal plug (considered as day 0.5 of gestation). Similarly, $Prx1Cre: Mir17-92^{fl/+}:Mir106b-25^{+/-}$ male and $Mir17-92^{fl/fl}:Mir106b-25^{-/-}$ female mice were mated to generate doubly conditional knockout mice. On the day 10.5 of gestation, pregnant female mice were sacrificed. The uterus was removed and washed in sterile calcium and magnesium-free PBS. E10.5 embryos were then recovered from the uterus and washed several times in PBS. Distal parts of fore and hind limb buds of embryos were isolated under a dissecting microscope using two pairs of sharp forceps and lysed in TRIzol (Molecular Research Center, Cincinnati, OH, USA). Total RNA was extracted using the Direct-Zol TM RNA MiniPrep kit (Zymo Research, USA) according to the manufacturer's instruction.

**RNA sequencing**. RNA-seq libraries were prepared with the Truseq mRNA kit (Illumina Inc., San Diego, CA, USA) with polyA + mRNA selection. Libraries were sequenced with the Illumina NextSeq system with 75 bp single end reads to generate 36–52 million reads per library. Reads were mapped to the mouse genome (mm10) with the STAR aligner[39], and differential expression analyses between biologic groups were performed with edger[39]. The significance level was set at $p <$ 0.05 (after correction for multiple hypothesis testing) for genes with mean FPKM >3 in at least one biologic group.

**qRT-PCR**. cDNA synthesis was performed using random hexamers using the DyNAmo cDNA Synthesis Kit (Finnzymes). Quantitative PCR was performed using the StepOnePlus Real-Time PCR System (Applied Biosystems) and the EvaGreen qRT-PCR mix (Solix BioDyne). Primer sequences are as follows: Pten-L, 5′-GAAAGGGACGGACTGGTGTA-3′ and Pten-R, 5′-AGTGCCACGGGTCTG-TAATC-3′; GAPDH-L, 5′-CACAATTTCCATCCCAGACC-3′ and GAPDH-R, 5′-GTGGGTGCAGCGAACTTTAT-3′; Smad2-L, 5′-AGTATGGACA-CAGGCTCTCC-3′ and Smad2-R, 5′-GTCTGCCTTCGGTATTCTG-3′; Smad4-L, 5′-ATCTGAGTCTAATGCTACCAGC-3′ and Smad4-R, 5′-TTCTTTGATGCTCTGTCTTGG-3′; Bim-L, 5′-GCCCCTACCTCCCTACAGAC-3′ and Bim-R, 5′- GGCATCACCGTGGATATTTT-3′; Tgfbr2-L, 5′-GCAGTGG-GAGAAGTAAAAGA-3′ and Tgfbr2-R, 5′- CCAGCCTGCCCCATAAGAGC-3′. Pik3r1-R, 5′-GCAGCTGAGTACCGAGAGAT-3′; Pik3r1-L, 5′-GCCACTCGTT-CAGCTTCTTC-3′; Pik3r3-R, 5′-ACCGAGTACAAGCAGAGGAC-3′; Pik3r3-L, 5′-GCAAAGCCATATCCTCGAGC-3′; Pik3r5-R, 5′-TTTCAGGGAAGGTGGC-TAGG-3′; Pik3r5-L, 5′-GTTCCGTGGCTTCTCTTCAC-3′; Pik3ip1-R, 5′-CAAA-GAGGCACAGGTGTTCC-3′; Pik3ip1-L, 5′-TGTAGCCCACGATAATGCCA-3′; Akt1s1-R, 5′-AAGAGGACAGAAGCCCGATC-3′; Akt1s1-L, 5′-GGAAGTCGCTGGTATTGAGC-3′. Expression of miR-17-92 miRNAs and miR-106b-25 miRNAs were determined using the mirVana qRT-PCR miRNA Detection Kit (Ambion).

**Luciferase reporter assay**. A 350-bp-long DNA fragment containing the conserved miR-17 binding sequence (5′-GCACTTTA-3′) or a mutated sequence (5′-GCTCTATA-3′) was PCR amplified using primers, Common R, 5′-GCAAGCTTCTCAGCTCCCTGGTCCATAA-3′ and Wt-F, 5′-GCACTAGT-GATCAGCTAATTGACCAGATGCACTTTATTAATGCCTGTGTGTAAA-TACGAA-3′ or Mut-F, 5′-GCACTAGTGATCAGCTAATTGACCAGATGCTCTA-TATTAATGCCTGTGTGTAAATACGAA-3′ and subcloned into pMIR-REPORT miRNA Expression Reporter Vector System (AM5795, ThermoFisher) at the Hind III and Spe I sites. miR-17 miRIDIAN microRNA mimic and a control mimic were purchased from Dharmacon. A renilla luciferase expression construct, a Tgfbr2-UTR reporter construct, and the miR-17 mimic or control mimic were co-transfected to primary limb bud cells isolated from E12.5 embryos using Attractene transfection reagent (Qiagen) according to manufacturer's instruction. Forty-eight hours after transfection, the cells were lysed, and luciferase and renilla luciferase activities were measured using the Dual-luciferase reporter assay system (Promega).

**Western blot analysis**. Anti-Smad2 (#5339), p-Smad2 (Ser465/467 #3108), p-Akt (Thr308, #2965), p-Akt (Ser473, #4060), NFkB (#3037), p-Erk (Ser259 #9911), p-p38 (#4511), p-Jnk (#4668), p-p38 (Ser) #9145), p-Stat3 (#9145), p-Stat1 (#9171), Tgfbr2 (#3713), p-IRS1 (Tyr895, 3070), p-Smad1/5(#9511), p-Pten (Ser380/Thr382/383, #9549), Pten (D4.3 #9188), and Act-b (13E5, #4970) antibodies were purchased from Cell Signaling Technology. Anti-p-Akt (T308) antibody (# 658320) was purchased from R&D Systems. Anti-Mycn (B8.4.B, #sc-53993) antibody was purchased from Santa Cruz Biotechnology. Anti-pCKIIβ (S209) antibody (STJ90892) was purchased from St Joh's laboratory. Anti-Gapdh antibody (10R-2932) was purchased from

Fitzgerald Industries International. Western blot analysis was performed according to the standard procedure. Uncropped blots are included in Supplemetary Figs. 9–12.

**Data availability**. The RNA-seq data are uploaded to the ArrayExpress database with the accession number, E-MTAB-5671 (https://www.ebi.ac.uk/arrayexpress/experiments/E-MTAB-5671/).

**Statistics**. Data are presented as mean ± SEM from indicated numbers of samples. For statistical analysis between the two groups, we used a 2-tailed Student's $t$ test; $p$ values less than 0.05 were considered statistically significant. Sample numbers were determined based on the effect size obtained during preliminary experiments to obtain a power of 0.8.

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

## Acknowledgements

This work is supported by the US NIH grant, AR056645 (TK). We thank Center for Skeletal Research (NIH P30 AR066261) for histological assistance. We thank Henry Kronenberg, Marc Wein, and Matthew Warman for valuable comments and suggestions.

## Author contributions

F.M. and T.K. conceived the project and designed the experiments. F.M., A.K., E.P., G.P., U.M.A., and T.K. performed the experiments and interpreted the results. F.M., A.K., and T.K. wrote the manuscript.

## Additional information

**Competing interests:** The authors declare no competing interests.

