## [Peer Review File · Nature Communications]

Editorial Note: This manuscript has been previously reviewed at another journal that is not operating a transparent peer review scheme. This document only contains reviewer comments and rebuttal letters for versions considered at Nature Communications. Mentions of prior referee reports have been redacted.

Reviewers' comments:

Reviewer #1 (Remarks to the Author):

The manuscript by Mirzamohammidi et al. investigates underlying mechanisms responsible for causing Feingold syndrome. Feingold Syndrome can be caused by haploinsufficiency of MYCN (Feingold syndrome type 1) or MIR17HG (Feingold syndrome type 2), with previous studies showing that miR-17-92 is a transcriptional target of MYCN. The authors show that conditional deletion of mir-17-92 with the Prx1-Cre driver, together with germline deletion of Mir106b-25, a paralogous gene encoding miR-17-92 family miRNAs, results in a phenotype consistent with type 2 disease and is similar to type 1 disease caused by Prx1-Cre deletion of MYCN. Although miR-17-92 is downregulated in MYCN-deleted limb bud mesenchyme as expected, the authors show that MYCN deficiency and miR17-92 deficiency do not act in a simple epistatic pathway to cause Feingold pathology, but instead cause disease through distinct mechanisms. In the type 2 model, the Tgf-beta receptor Tgfr2 was increased, TGF-beta signaling was upregulated, and Tgfr2 heterozygosity or small molecule inhibition of TGF-beta receptors during a period of early skeletal mesenchyme development was effective at rescuing disease pathology. In contrast, there was no upregulation of TGF-beta signaling in the MYCN conditional knockout model and instead there was a diminution of Akt signaling and PTEN heterozygous deletion rescued the MYCN cKO phenotype.

General Comments

This is a fascinating and important study that, through genetic and small molecule inhibitor "rescue" experiments, clearly demonstrate the mechanistic role of Tgf beta and PTEN/AKT signaling in causing type 2 and type 1 Feingold syndrome models. What is particularly confusing however, is the odd way that PTEN/AKT signaling was found to be responsible for the MYCN/type 1 disease phenotype. Whereas the type 2 model caused by deletion of miR-17-92 was found to be due to excessive Tgf beta signaling (and not upregulation of the previously identified 17-92 target Pten), transgenic overexpression of miR-17-92 had no effect on Tgf signaling, but it rescued the type 2/NMYC cKO phenotype through the apparent suppression of Pten and increased Akt signaling. Confusion arises because the latter rescue appears to be the result of potentially spurious inhibition of Pten and increased AKT activity caused by "supraphysiological levels of miR-17-92" – which in turn led to the genetic rescue experiments showing PTEN deficiency rescues the MYCN cKO/type 1 phenotype. PTEN is a previously described target of miR17-92 as pointed out earlier in the manuscript. I suggest that the model in Figure 6e show that miR-17-92 regulates Pten (when overexpressed), and that text be added describing/discussing the differential and confusing regulation of Pten by N-Myc and miR17-92 and the differential regulation of Tgf beta and Pten signaling by miR17-92.

A further discussion of the discrepancy in mechanism between the author's work and that proposed by de Pontual et al. would also be useful.

Specific Comments

1. Figure 1e. It could bring more clarity to the report if E14.5 expression of 17-92 miRNAs was shown in comparison to limb buds at E10.5 and E11.5 embryos (MYCN cKO and control), instead of having the latter data presented in Fig. 5e. Establishing that miR17-92 is not much changed at E10.5 and knowing this early may help the flow of the manuscript and avoid confusion.
2. Figure 1e. It should be more explicitly stated in the text or Figure legend that the miRNAs

examined are all expressed from the miR-17-92 gene cluster since the nomenclature can be extremely confusing.

3. Figure 1e. It is stated that the cells used are isolated from E14.5 limb buds. E14.5 is a post limb bud stage where the skeletal elements are present and digits have separated. The text, as well as the Methods section on isolation of limb bud cells and the reference cited (which does necessarily apply to E14.5, should be amended accordingly.

4. It would be useful to show the phenotype, if any, of 17 tg mice. A demonstration of the effects miR17-92 overexpression in the TG mice on Tgf-beta and Pten/Akt signaling could be useful in further confirming the conclusions reached.

5. In Figure 5e, it is shown that miR17-92 is most downregulated at E14.5. In contrast to widespread NMyC expression in limb bud mesenchyme, at E14.5, digits are separated and N-Myc expression is restricted to perichondral regions. The data presented suggest that miR17-92 is not a NMyC target gene and it may be worth noting that the downregulation observed at E14.5 in the absence of NMyC could simply reflect the loss of undifferentiated interdigital mesenchyme that express these miRNAs. Proof would require in situ hybridization of the miRNAs in control and MYCN cko mice.

6. Did the authors test whether PTEN heterozygosity rescues the 17dKO phenotype? The data suggests that there would be no rescue. Such information would be highly relevant to the conclusions reached.

7. For the presentation of mouse phenotypes in Figures 5 and 6, the number of mice analysed should be indicated in the text or in the figure legends.

Minor

On line 167 it is stated that "Casein kinase 2 (CK2) is a major regulator of Pten phosphorylation and activity, and CK2 and Myc have been shown to synergistically promote oncogenesis. It may be worth noting that the reference cited suggests that the synergy is due to CK2 regulation of Myc stability.

Line 389 – A set of references are misplaced here

There are a number of typographical errors throughout

The layout of figure 5 could be reconfigured to better reflect the order of data presented.

Reviewer #2 (Remarks to the Author):

Review of "Mouse models of Feingold Syndrome type 1 and type 2 reveal distinct molecular pathways mediating actions of Mycn and Myc-regulated miR-17~92 microRNAs", by Mirzamohammadi and colleagues.

In this interesting manuscript, the authors have used a combination of mouse genetics and pharmacologic methods to investigate the molecular mechanisms through which loss of function mutations in Mycn and miR-17~92 promote skeletal defects. Germline heterozygous mutation of MYCN or monoallelic deletion of the miR-17~92 are observed in patients affected by Feingold Syndrome 1 and 2, respectively. Because miR-17~92 is a direct target of MYCN, it has been hypothesized that miR-17~92 acts epistatically to MYCN in the pathogenesis of Feingold Syndrome. In support of this hypothesis, the authors confirm that loss of Mycn leads to

concomitant reduction of miR-17~92 in mesenchymal cells in vivo and show for the first time that ectopic expression of miR-17~92 can rescue the skeletal defects caused by loss of MYCN. A more in depth analysis, however, reveals that while the skeletal consequences of miR-17~92 inactivation can be rescued by pharmacologic or genetic inhibition of Tgfrb2 signaling, the skeletal phenotypes caused by Mycn deletion cannot be rescued. Rather, the authors show that inhibition of PI3K signaling can rescue skeletal development in Mycn-mutant mice. To explain this apparent discrepancy, the authors propose that Mycn and miR-17~92 act through independent pathways to modulate skeletal development and suggest that the rescue of the skeletal defects in Mycn-null/miR-17~92 overexpressing mice is due to inhibition of Pten by supra-physiologic levels of miR-17~92.

The manuscript is well written, the experiments are clearly described, and the results are of significant interest to scientific community. I have, a few comments and suggestions that I would like to see addressed before publication. More specifically:

a) The authors should make clear that while conditional homozygous deletion of miR-17~92 (with or without concomitant inactivation of miR-106b~25) or Mycn in the limb buds and skull mesenchyme of developing embryo is an excellent approach to study the role of these genes in skeletal development, it is NOT a model of Feingold Syndrome. In fact, a true model of Feingold Syndrome is germline monoallelic deletion of either genes. In this context, it would be important to determine whether heterozygous loss of Tgfrb2 (or pharmacologic inhibition of TGFb signaling) can rescue the skeletal defects observed in miR-17~92^{+/-} and Mycn^{+/-} mice, respectively.

b) With respect to the rescue experiments, the authors should complement the representative images shown in figures 5 and 6 with quantitative data (including statistical analysis) and specify how many animals were analyzed for each genotype/condition.

c) A 2015 Nature Genetics paper has shown that the skeletal defects caused by loss of miR-17~92 are almost exclusively due to loss of miR-17 and miR-20a (PMID: 26029871). The authors should determine whether Tgfrb2 is a direct target of these two miRNAs in mesenchymal cells. According to Targetscan 7, there is a single perfect miR-17 binding site (8-mer) in the 3'UTR of Tgfrb2. The ideal experiment would be to mutate this site using CRISPR, but at a minimum a simple luciferase reporter experiment using the Tgfrb2 3'UTR should be included in the revised manuscript.

Authors' Response to the Review Comments:

We appreciate reviewers' careful reading of our manuscript and thoughtful comments, suggestions and criticisms. We performed additional experiments along the reviewers' questions and suggestions. We believe that we have significantly improved our manuscript and hope that now the quality of the paper is satisfactory for *Nature Communications*. Please find our point-by-point response as below.

Reviewer #1 (Remarks to the Author):

The manuscript by Mirzamohammidi et al. investigates underlying mechanisms responsible for causing Feingold syndrome. Feingold Syndrome can be caused by haploinsufficiency of MYCN (Feingold syndrome type 1) or MIR17HG (Feingold syndrome type 2), with previous studies showing that miR-17-92 is a transcriptional target of MYCN. The authors show that conditional deletion of mir-17-92 with the Prx1-Cre driver, together with germline deletion of Mir106b-25, a paralogous gene encoding miR-17-92 family miRNAs, results in a phenotype consistent with type 2 disease and is similar to type 1 disease caused by Prx1-Cre deletion of MYCN. Although miR-17-92 is downregulated in MYCN-deleted limb bud mesenchyme as expected, the authors show that MYCN deficiency and miR17-92 deficiency do not act in a simple epistatic pathway to cause Feingold pathology, but instead cause disease through distinct mechanisms. In the type 2 model, the Tgf-beta receptor Tgfr2 was increased, TGF-beta signaling was upregulated, and Tgfr2 heterozygosity or small molecule inhibition of TGF-beta receptors during a period of early skeletal mesenchyme development was effective at rescuing disease pathology. In contrast, there was no upregulation of TGF-beta signaling in the MYCN conditional knockout model and instead there was a diminution of Akt signaling and PTEN heterozygous deletion rescued the MYCN cKO phenotype.

General Comments

This is a fascinating and important study that, through genetic and small molecule inhibitor "rescue" experiments, clearly demonstrate the mechanistic role of Tgf beta and PTEN/AKT signaling in causing type 2 and type 1 Feingold syndrome models. What is particularly confusing however, is the odd way that PTEN/AKT signaling was found to be responsible for the MYCN/type 1 disease phenotype. Whereas the type 2 model caused by deletion of miR-17-92 was found to be due to excessive Tgf beta signaling (and not upregulation of the previously identified 17-92 target Pten), transgenic overexpression of miR-17-92 had no effect on Tgf signaling, but it rescued the type 2/NMYC cko phenotype through the apparent suppression of Pten and increased Akt signaling. Confusion arises because the latter rescue appears to be the result of potentially spurious inhibition of Pten and increased AKT activity caused by "supraphysiological levels of miR-17-92" – which in turn led to the genetic rescue experiments showing PTEN deficiency rescues the MYCN cko/type 1

phenotype. PTEN is a previously described target of miR17-92 as pointed out earlier in the manuscript. I suggest that the model in Figure 6e show that miR-17-92 regulates Pten (when overexpressed), and that text be added describing/discussing the differential and confusing regulation of Pten by N-Myc and miR17-92 and the differential regulation of Tgf beta and Pten signaling by miR17-92.

Response) Thank you for the supportive comments on our manuscript and valuable suggestions. We have revised the figure and discussion as suggested.

A further discussion of the discrepancy in mechanism between the author's work and that proposed by de Pontual et al. would also be useful.

Response) The previous study and ours both found that the *Mycn* deletion resulted in partial reductions in miR-17-92 expression. In our study, we found that the magnitude of the miR-17-92 reduction in *Mycn*-deficient limbs was not large enough to cause a significant TGF- β signaling deregulation during early limb development. We have included this discussion.

Specific Comments

1. Figure 1e. It could bring more clarity to the report if E14.5 expression of 17-92 miRNAs was shown in comparison to limb buds at E10.5 and E11.5 embryos (MYCN cko and control), instead of having the latter data presented in Fig. 5e. Establishing that miR17-92 is not much changed at E10.5 and knowing this early may help the flow of the manuscript and avoid confusion.

Response) Thank you for your suggestion. We have replaced Fig. 1e by 5e.

2. Figure 1e. It should be more explicitly stated in the text or Figure legend that the miRNAs examined are all expressed from the miR-17-92 gene cluster since the nomenclature can be extremely confusing.

Response) We have indicated that these miRNAs are encoded in the miR-17-92 cluster miRNA gene in the legend.

3. Figure 1e. It is stated that the cells used are isolated from E14.5 limb buds. E14.5 is a post limb bud stage where the skeletal elements are present and digits have separated. The text, as well as the Methods section on isolation of limb bud cells and the reference cited (which does necessarily apply to E14.5, should be amended accordingly.

Response) We have included the procedure of RNA extraction from limb buds and limbs in the methods section; four limb buds of E10.5 or E11.5 embryos

were dissected and directly lysed in Trizol. For E14.5 limbs, paws of four limbs were skinned before lysing in Trizol.

4. It would be useful to show the phenotype, if any, of 17 tg mice. A demonstration of the effects miR17-92 overexpression in the TG mice on Tgf-beta and Pten/Akt signaling could be useful in further confirming the conclusions reached.

Response) miR17-92 overexpression (*Mir17-Tg*) alone causes limb overgrowth. We have described this phenotype in the main text and included data showing the digit phenotype of *Mir17-Tg* mice in Supplemental Fig.3.

5. In Figure 5e, it is shown that miR17-92 is most downregulated at E14.5. In contrast to widespread NMyC expression in limb bud mesenchyme, at E14.5, digits are separated and N-Myc expression is restricted to perichondral regions. The data presented suggest that miR17-92 is not a NMyC target gene and it may be worth noting that the downregulation observed at E14.5 in the absence of NMyC could simply reflect the loss of undifferentiated intergital mesenchyme that express these miRNAs. Proof would require in situ hybridization of the miRNAs in control and MYCN cko mice.

Response) We have performed *In situ* hybridization (ISH) to detect mature miR-17-5p miRNA on E14.5 limb sections using a LNA oligo probe.

Based on examination on H/E-stained sections of the forelimb, *Mycn* conditional KO limbs show delay in cartilage maturation, but otherwise there are no significant changes in cellular makeup. At this stage, mature miR-17 (mmu-miR-17-5p) is expressed mainly in chondrocytes and perichondrial cells. miR-17 expression appears to be generally reduced in *Mycn*-deficient limbs including chondrocytes. It is unclear why miR-17 expression in chondrocytes is reduced in *Mycn* cKO mice. Since the half-life of mature miRNAs is very long, it may be a consequence of reduced miR-17 expression in the prechondrogenic stage.

Nevertheless, we did not observe significant changes in cellular makeup in *Mycn* cKO limbs at E14.5, and our result (reduced miR-17-92 expression in E14.5 *Mycn* cKO limbs, determined by qRT-PCR and ISH) is consistent with the previously reported result (de Pontual et al. Nat Genet 2011; Ref. 2).

This piece of data is now included in Supplementary Fig 2.

6. Did the authors test whether PTEN heterozygosity rescues the 17dKO phenotype? The data suggests that there would be no rescue. Such information would be highly relevant to the conclusions reached.

Response) Yes, we tested whether *Pten* heterozygosity could rescue the 17dKO phenotype. *Pten* heterozygosity was not able to rescue the brachysyndactyly phenotype of 17dKO mice. This result is included in Supplementary Fig. 8d.

7. For the presentation of mouse phenotypes in Figures 5 and 6, the number of mice analysed should be indicated in the text or in the figure legends.

Response)

For Fig. 5, We confirmed the reproducibility of the results (failed rescue of the brachysyndactyly phenotype of *Mycn* cKO mice by *Tgfb2* heterozygosity) in more than 3 compound mutants. This statement is included in the legend. For Fig. 6 showing the successful partial rescue of the *Mycn* cKO phenotype by *Pten* heterozygosity, we have included the sample number and measurement results in the legend.

Minor

On line 167 it is stated that “Casein kinase 2 (CK2) is a major regulator of Pten phosphorylation and activity, and CK2 and Myc have been shown to synergistically promote oncogenesis. It may be worth noting that the reference cited suggests that the synergy is due to CK2 regulation of Myc stability.

Response) We have included the statement. Thank you.

Line 389 – A set of references are misplaced here

Response) We have corrected the reference location.

There are a number of typographical errors throughout

Response) Thank you for pointing these out. We have thoroughly checked the manuscript and corrected typographical errors that were found.

The layout of figure 5 could be reconfigured to better reflect the order of data presented.

Response) We have reconfigured Fig. 5.

Reviewer #2 (Remarks to the Author):

Review of "Mouse models of Feingold Syndrome type 1 and type 2 reveal distinct molecular pathways mediating actions of Mycn and Myc-regulated miR-17~92 microRNAs", by Mirzamohammadi and colleagues.

In this interesting manuscript, the authors have used a combination of mouse

genetics and pharmacologic methods to investigate the molecular mechanisms through which loss of function mutations in Mycn and miR-17~92 promote skeletal defects. Germline heterozygous mutation of MYCN or monoallelic deletion of the miR-17~92 are observed in patients affected by Feingold Syndrome 1 and 2, respectively. Because miR-17~92 is a direct target of MYCN, it has been hypothesized that miR-17~92 acts epistatically to MYCN in the pathogenesis of Feingold Syndrome. In support of this hypothesis, the authors confirm that loss of Mycn leads to concomitant reduction of miR-17~92 in mesenchymal cells in vivo and show for the first time that ectopic expression of miR-17~92 can rescue the skeletal defects caused by loss of MYCN. A more in depth analysis, however, reveals that while the skeletal consequences of miR-17~92 inactivation can be rescued by pharmacologic or genetic inhibition of Tgfrb2

signaling, the skeletal phenotypes caused by Mycn deletion cannot be rescued. Rather, the authors show that inhibition of PI3K signaling can rescue skeletal development in Mycn-mutant mice. To explain this apparent discrepancy, the authors propose that Mycn and miR-17~92 act through independent pathways to modulate skeletal development and suggest that the rescue of the skeletal defects in Mycn-null/miR-17~92 overexpressing mice is due to inhibition of Pten by supra-physiologic levels of miR-17~92.

The manuscript is well written, the experiments are clearly described, and the results are of significant interest to scientific community. I have, a few comments and suggestions that I would like to see addressed before publication. More specifically:

Response) We appreciate Reviewer #2's overall positive comments and suggestions to improve our manuscript.

a) The authors should make clear that while conditional homozygous deletion of miR-17~92(with or without concomitant inactivation of miR-106b~25) or Mycn in the limb buds and skull mesenchyme of developing embryo is an excellent approach to study the role of these genes in skeletal development, it is NOT a model of Feingold Syndrome. In fact, a true model of Feingold Syndrome is germline monoallelic deletion of either genes. In this context, it would be important to determine whether heterozygous loss of Tgfrb2 (or pharmacologic inhibition of TGFb signaling) can rescue the skeletal defects observed in miR-17~92+/- and Mycn+/- mice, respectively.

Response) We agree with the reviewer's concern that our models are not genetic equivalents of human syndromes. As explained in the text, we used these models (conditional homozygous knockouts) primarily because heterozygous loss does not fully recapitulate the condition of the human patients. We believe that our mouse models share the same pathological mechanisms with human diseases because 1) these mice show skeletal abnormalities (synbrachydactyly) similar to patients with these syndromes, and 2) there is a

direct correlation of the severity of the phenotype and the number of allelic loss of *Mir17-92/106b* genes in mutant mice.

In this revision, to address this reviewers' concern, we have generated germline heterozygous mutant mice for *Mycn* and *Mir17-92* genes. Unlike human patients, germline heterozygous *Mir17-92(+/-)* mice show only modest reductions in body size and the second phalange of the fifth digit. In our hands, the shortening of the second phalange of *Mir17-92(+/-)* mice is very mild and variable (Please see the Figure below). We also generated germline *Mir17-92(+/-)* mice in the background of *Mir106b-25(-/-)*, hoping to obtain an analyzable phenotype. Even these compound heterozygotes [*Mir17-92(+/-):Mir106b-25(-/-)*] did not show a phenotype strong enough for quantitative analysis (Right panel of the Figure below).

Therefore, it was not possible for us to reliably assess the rescue effect in these models due to the modesty of the basal phenotype.

In our hands, we have found that the digit phenotype of conditional heterozygous deletion is less variable and slightly more severe than germline heterozygotes. Therefore, as an alternative, we have performed rescue experiments using a TGF- β receptor inhibitor (LY364947) on conditional heterozygous knockout mice (Supplementary Fig. 1 and 6) to test whether the phenotype of heterozygous deletion is also caused by the same mechanism as in homozygous knockouts. As shown in the Supplementary Fig. 6a, we confirmed that TGF- β inhibitor injection from E9.5 through E18.5 rescued the shortening the second phalanx in *Mir17-92* heterozygotes. We have included this result.

As for *Mycn(+/-)* mice, we have generated germline *Mycn* heterozygotes by deleting the conditional allele in germ cells. As shown in the picture below, heterozygous *Mycn (+/-)* mice show no detectable digit phenotypes (neither brachydactyly nor syndactyly), and therefore, it was not possible to perform rescue experiment using these mice.

We speculate that the dosage requirement of *Mycn* and *Mir17-92* for normal limb development is different between humans and mice; we have included this statement in discussion section.

Figure Absence of brachysyndactyly in *Mycn*(+/-) mice at 1 month of age.

b) With respect to the rescue experiments, the authors should complement the representative images shown in figures 5 and 6 with quantitative data (including statistical analysis) and specify how many animals were analyzed for each genotype/condition.

Thank you for your suggestion. We have confirmed the reproducibility of the effects of genetic rescue in each model by analyzing 3 - 5 rescued animals (the number in each experiment is indicated in legends). As for the failed rescue experiment of *Mycn* cKO mice by *Tgfr2* heterozygous deletion in Fig. 5, unfortunately we were unable to perform measurements because the skeletal prep samples degraded. Nevertheless, as indicated, we confirmed that *Tgfr2* deletion did not rescue the brachysyndactyly of *Mycn* cKO mice by analyzing more than three compound mutant mice. For Fig. 6, we have now included measurements of metatarsal bones and indicated the sample number of each model, and performed statistical analysis. This result demonstrates that the reduced size in bone is effectively rescued by *Pten* heterozygosity, although the

rescue is not complete as fused bones and diminished phalanxes still remain. This is stated in the text.

c) A 2015 Nature Genetics paper has shown that the skeletal defects caused by loss of miR-17~92 are almost exclusively due to loss of miR-17 and miR-20a (PMID: 26029871). The authors should determine whether Tgfr2 is a direct target of these two miRNAs in mesenchymal cells. According to Targetscan 7, there is a single perfect miR-17 binding site (8-mer) in the 3'UTR of Tgfr2. The ideal experiment would be to mutate this site using CRISPR, but at a minimum a simple luciferase reporter experiment using the Tgfr2 3'UTR should be included in the revised manuscript.

Response) Thank you for the suggestion. Accordingly, we have tested whether miR-17-5p could suppress gene expression through the predicted binding site of *Tgfr2* by luciferase reporter assay. We have confirmed that miR-17-5p suppressed luciferase expression in a sequence-specific manner. These results are now included in Fig. 3e. We have also cited this paper in the text.

We appreciate your time for evaluating our manuscript.

Reviewers' Comments:

Reviewer #1:

Remarks to the Author:

The authors have made substantive, constructive changes and additional data that have improved the impressive body of work in this manuscript. I feel the manuscript is now suitable for publication.

Reviewer #2:

Remarks to the Author:

The authors have satisfactorily addressed my concerns and I congratulate them for an interesting and well written manuscript.

Authors' Response to the Review Comments:

Thank you for the opportunity to revise our manuscript.

We therefore invite you to revise your paper one last time to address the remaining concerns of our reviewers. In particular, we consulted with referee 1 on the issues with figure substitutions that you highlighted. This referee comments that:

1) in figure 5c, the blots show a trend that appears consistent with your conclusion, and suggests that quantification should be included.

Response) Quantification of pSmad2/tSmad2 and pAkt/tAkt, calculated by band intensity, is included beneath the blot picture in Fig. 5c.

2) For Fig. 6, to confirm that the the low-exposure p-Pten blot does not compare with the adjacent high exposure blot. Could you please confirm that you are presenting here low and high-exposure blots. Also please include the full blots in the supplementary figures (together with the other uncropped gel scans), and to show 2 lanes in figure 6. Also please ensure that you upload the correct amended figures.

Response) The last two lanes (Ctrl and cKO) of the long-exposed image were cropped and used for p-Pten presentation in the figure, and therefore control and cKO bands were compared in the same blot. The short exposed image is for Actb. The full size gel images with short and long exposure are included in Supplementary Fig. 11.

At the same time we ask that you edit your manuscript to comply with our format requirements and to maximise the accessibility and therefore the impact of your work.

We hope that the revised manuscript is satisfactory.
Thank you for your time for reviewing our work.